# Vehicle-to-Grid Market Readiness in Europe with a Special Focus on Germany

Christopher Hecht [1,2,3,4,*] , Jan Figgener [1,3,4] and Dirk Uwe Sauer [1,3,4,5,*]

1. Grid Integration and Storage System Analysis, Institute for Power Electronics and Electrical Drives (ISEA), RWTH Aachen University, 52074 Aachen, Germany
2. The Mobility House, St.-Martin-Straße 57, 81669 München, Germany
3. Institute for Power Generation and Storage Systems (PGS), E.ON ERC, RWTH Aachen University, 52074 Aachen, Germany
4. Juelich Aachen Research Alliance, JARA-Energy, 52056 Aachen, Germany
5. Helmholtz Institute Muenster (HI MS), IEK-12, Forschungszentrum Jülich, 52428 Jülich, Germany
* Correspondence: christopher.hecht@isea.rwth-aachen.de (C.H.); batteries@isea.rwth-aachen.de (D.U.S.)

**Abstract:** Vehicle-to-grid means that electric vehicles are charged when electricity is plentiful and discharged when it is scarce. New battery-electric vehicles have an energy capacity above 60 kWh installed and practically always have a DC connector. With over 1 million of such vehicles in Germany alone already, the flexibility potential to balance out fluctuating renewable generation or compensate for grid constraints is large. While many actors are working to enable this market, the readiness of hardware and regulations as well as the potential volume are hard to grasp. This paper provides an overview of these factors for Europe with a special focus on Germany. We find that some countries started to implement regulatory frameworks but none are ready yet. Issues include taxation, the fulfillment of grid codes, and the lack of smart meters. In terms of vehicles, 25 manufacturers with bidirectional charging ability were identified, but most vehicles were only used in field tests or operate in island mode. In terms of charging infrastructure, the picture is brighter with at least 20 manufacturers that offer DC bidirectional charging stations and 2 offering an AC variant.

**Keywords:** vehicle-to-grid; electric vehicles; market readiness; bidirectional charging





## 1. Introduction

Vehicle-to-grid (V2G) describes the process where electric vehicles (EVs) not only charge electricity from the grid but can also discharge. In this way, vehicle batteries can be used to stabilize the electricity grid if excess or insufficient power generation is available or if power lines are overloaded and cannot transport electricity from the point of generation to consumers. In doing so, the technology promises to address a key challenge of the energy transition relying on (non-dispatchable) renewable energy sources: they cannot control their power generation according to the electricity demand but have to rely on energy storage units and flexible consumers to match the electricity supply and demand. EVs can offer this flexibility by charging when electricity is abundant and discharging when it is scarce. While the concept has extensively been discussed in the literature, real-world applications of the concept have been largely limited to trial projects and small demonstrators [1–4]. In recent times, however, various actors in industry have launched products and services that implement the bidirectional charging of EVs.

With the term vehicle-to-grid first being coined in 2001 [5], these developments have been closely monitored within academia and industry from various points of view. One of the more common ones is to determine the economics of such a system. In 2017, Steward [6] found that studies strongly diverged in their expected profitability where the net revenue for light-duty vehicles varied between USD 4/a and USD 3320/a. Varying assumptions and markets that vehicles participate in can explain the wide range of values. Heilmann

and Friedl [7] showed that these assumptions and markets can be grouped into four overarching categories, namely market conditions, technical conditions, costs, and controls. They further confirm that the range of revenues and potential business cases is wide with some cases incurring losses of up to EUR 700 /a while other studies found revenues of EUR 5200. Another way to categorize the applications of vehicle-to-grid is to look at the exact type of service that they provide to the grid as Ravi and Aziz [8] have carried out. They highlight that vehicle-to-grid is only one of many aspects of the broader category of vehicle-grid integration which also encompasses smart charging. In practice, however, the term vehicle-to-grid has become almost synonymous with vehicle-grid integration, which the authors also confirm. Next to these more general analyses, researchers have started to investigate topics that are more specific as well. Examples are works focusing on specific countries such as Indonesia [9], China [10,11], Germany [12], the Netherlands [13], and many more.

The economic analysis has another side to it—the marketing of the concept of V2G to potential users. Baumgartner et al. [14] followed a survey-based approach and asked people with low, medium, and high levels of experience with EVs to participate. They found that a minimum remaining range in the vehicle as well as in climate-neutral charging are key instruments to motivate people, even if this results in lower financial compensation. Bohdanowicz et al. [15] show an even more diverse motivation landscape and find that the size of financial rewards does not show a relationship with the willingness to participate in V2G schemes. They argue that intrinsic and altruistic motivations are highly important.

Another path that is frequently taken is to check how vehicle-to-grid operation affects battery life. Lehtola et al. published an early review as early as in 2019 [16] that indicated what subsequent studies have frequently confirmed: if the operation considers battery aging, smart and bidirectional charging can reduce aging relative to charging a vehicle directly after arrival to a full state of charge. Calearo and Marinelli show that additional aging from extensive V2G operation over 5 years yields a positive net profit of appr. EUR 3500 per vehicle over the given period [17,18].

With the ongoing developments and announcements, it can be hard to obtain an overview of the current state of the market. Market reports [19–22] and research papers [23] largely focus on the electric vehicle market as a whole but often address the topic of V2G as only a side topic. For research that does focus on the V2G domain, a challenge is staying up to date as the market moves quite dynamically. Parts of what Das et al. [24] considered a future development is already being developed only three years later. With this paper, we aim to complement the existing literature by providing an analysis dedicated to V2G and related topics including the latest developments in markets, regulation, and technology. Our geographical focus lies on Europe for an analysis of the readiness of the regulatory body as well as the available bidirectional vehicles and charging stations. A further deep dive with regard to market potential is executed for Germany, the leading car market in Europe in terms of the number of vehicles sold. In doing so, we hope that we aid both practitioners and researchers by providing clarity regarding the available capacity and technologies both now and—by extrapolation—in the coming years. Particularly, actors looking at raw material demand, manufacturing volumes, and grid services require a realistic picture of the current market dynamic, which we hope to provide with this submission.

## 2. Materials and Methods

This paper is based on two main pieces of work—extensive research to gather information about the readiness of regulatory frameworks, vehicles, and chargers as well as data analysis for a market analysis of Germany.

### 2.1. Literature Review

The literature review performed for the purpose of this paper was performed in April 2023 by The Mobility House. The main data sources are product catalogues and reports on field tests. Given the nature of the data collection method, the list of car and charging

station models may be incomplete and the evaluation of the regulatory state by country contains partially subjective evaluations. Both do not claim completeness and with the fast-changing market, the situation may change rapidly.

*2.2. Data Analysis*

The data analysis performed in this paper is based on data collected previously as reported in [23] and available in an interactive format in [25]. The work was performed by RWTH Aachen. For the sake of conciseness, the data collection method is only reported in brief in this paper and the reader can refer to [23].

For the years observed, the car registrations were retrieved from the German Federal Office for Motor Traffic (in Flensburg, Germany, "Kraftfahrtbundesamt", KBA) [26] and merged with a catalogue containing technical car data from the General German Automobile Association ("Allgemeiner Deutscher Automobil Club", ADAC) [27]. Based on this approach, the battery capacity, charging power, and DC fast-charging connectors could be determined for most car models. This error range is created because of the fact that the KBA does not assign model identifiers if vehicles do not differ strongly. Since the number of vehicles that we could assign technical properties to does not exactly match the actual number of registered battery electric vehicles (BEVs) and of plug-in hybrid electric vehicles (PHEVs), scaling was performed for each datapoint reported. Since the plots shown in this paper rely on different primary data sources, we report the precise data sources in the caption of each plot separately.

## 3. Results

The results presented in this paper are grouped into four sections. The first two, "overview of V2G-readiness of the regulatory framework" and "Overview of the V2G-readiness by hardware supplier", focus on what is currently possible in terms of regulation, vehicles, and charging stations. Most devices are still at an early stage and have not yet established themselves strongly in the market. Additionally, many of the vehicles that we could identify as bidirectional are either only able to supply loads in a vehicle-to-load setup or only support V2G in field tests. Neither of those two options constitutes a V2G-ready solution for customers. Therefore, the latter two sections, "Electric Vehicle Sales in Germany" and "Properties of BEVs Sold in Germany" focus on the entire fleet of BEVs in Germany and not only on vehicles with the ability to charge bidirectionally. To support the topic of this paper, the analysis of the current fleet is conducted with a focus on what effect and implications the analyzed vehicle properties would have for a V2G operation.

*3.1. Overview of the V2G Readiness of the Regulatory Framework*

For V2G applications to be possible, having a technical solution available is insufficient. In addition, the regulatory environment needs to be such that vehicles are able to interact with the grid and the electricity market. Focusing on Europe, the main hurdles for such a market adoption are as follows:

- Taxation as consumption In many European countries, electricity consumption is measured solely in kWh and significant taxes and levies are applied to the energy consumed. In Germany, for instance, the electricity price has been approximately EUR 30 ct/kWh for the last decade [28] while wholesale power prices were around EUR 4 to 5 ct/kWh [29]. This means that it was virtually impossible that margins were high enough that selling the heavily taxed electricity back to the grid was economically sensible. With the energy crisis starting in 2021, this ratio has somewhat changed as higher price fluctuations became possible, but without changes in market rules, it was still near impossible to create a business case. A way to overcome this issue is by reimbursing taxes and levies in a way that is proportional to the amount of electricity sold back to the grid. This approach is currently pursued in Germany and many other countries in Europe. In this way, arbitrage trading becomes possible.

- Fulfillment of grid codes In many legislations, grid codes were designed for (large) stationary power producers and the concept of an asset that can feed electricity back into the grid at various locations was not foreseen. Part of the challenge is that a power generator's behavior in the case of short-circuits, voltage drops, or other irregular grid events is not uniform across distribution grids in Europe. For DC bidirectional charging, this is not a big problem since the grid codes can be supplied to the DC/AC converter inside the DC charging station. If the vehicle were to use AC bidirectional charging, however, the vehicle would have to alter its behavior based on location. Although ISO 15118-20 [30] allows for this behavior, few charging stations are actually able to implement this feature at the time of writing. The commonly chosen approach is to limit the bidirectional charging of a vehicle to a single charging station [31]. For details on this issue, please see Appendix A.

- Measurement of delivered energy Part of market interaction is the measurement and billing of delivered energy. To achieve this, smart meters are typically used. The deployment of these devices is vastly different across Europe with the Nordics, Italy, and France, and some smaller countries in the lead with approx. 90% or more households having such a device installed [32]. Others such as the United Kingdom (49% at the end of 2021 [33]) or Germany (0% at the end of 2021 [33]) lag far behind. Germany has realized this shortcoming and with the new Smart Meter Law [34], households have the right to have a smart meter installed for a maximum price of EUR 20 p.a. and one can be optimistic that many of the laggards in Europe will improve in this aspect. Without a smart meter, an economic operation is often limited to behind-the-meter options such as optimizing the self-consumption of onsite PV generators, but systematic benefits are harder to capture.

Table 1 shows an overview of how far each country in Europe is in the regulatory adoption of V2G. While many relevant markets have made efforts to adopt regulations to make V2G possible in a market environment, no country at the time of writing had a full integration in place yet.

**Table 1.** The state of the regulatory adoption of V2G in the 10 countries with the highest number of BEVs sold in 2022. For details on specific countries, see [35]. Data were gathered in April 2023.

| Country | State of Regulatory Affairs | |
|---|---|---|
| | Not Yet Begun | Rules under Development |
| Germany | | X |
| United Kingdom | | X |
| France | | X |
| Norway | X | |
| Sweden | X | |
| Netherlands | | X |
| Italy | X | |
| Belgium | X | |
| Switzerland | | X |
| Austria | X | |

Note that the statements made in this chapter are largely independent of the type of market that the vehicle interacts with. Since the gravity of the existing issues affects different markets differently, a short overview over the relevant two markets—frequency restoration markets and wholesale electricity markets—and the corresponding challenges is given below in Table 2. We do not include other service markets such as black start, capacity markets, redispatch, etc., as those are secondary in terms of volume to the discussed ones.

**Table 2.** Gravity of the stated regulatory challenges for the two key markets of frequency restauration services and wholesale.

| Market Challenge | Frequency restauration service markets | Wholesale electricity market |
|---|---|---|
| Taxation as consumption | Since the volume of energy delivered/consumed is typically low, this issue is secondary. | Problematic since large volumes of electricity are traded |
| Fulfillment of grid codes | Equally challenging as it is a prerequisite for market interactions | |
| Measurement of delivered energy | Meters need to measure energy delivered on the second level | Energy delivered per quarter-hour with sufficient accuracy |

*3.2. Overview of V2G Readiness of Hardware Supplier*

There is a strong market dynamic with more and more manufacturers moving into the V2G space, both with respect to vehicles as well as charging stations. Table 3 provides an overview of which vehicles are able to charge bidirectionally and Table 4 provides the same for charging stations. Note that data collection was in April 2023 and newer products may have been announced since. Further, note that bidirectional charging does not necessary mean that a vehicle or a station is also able to perform this service in the off-the-shelf configuration. Especially for the list of vehicles, limitations apply: many vehicles listed are able to only supply power in a vehicle-to-load mode where they are not connected to the power grid (e.g., Ioniq 5 and 6, BYD Atto3, Nio EL7, etc.) while others have only been tested in the V2G mode (e.g., Fiat 500e; BMW i3). The list of truly V2G-enabled vehicles is much shorter. Vehicles using the CHAdeMO plug such as e-NV200, Leaf, i-MiEV, or Outlander have already been able to discharge into the grid for years while the list of CCS-based vehicles with bidirectional capability is much shorter, including F-150 Lighting, ID.Buzz, or EX90 (announced). Further, note that the response speed of V2G-enabled hardware is partially too slow to participate in highly dynamic markets such as frequency containment reserve markets.

**Table 3.** List of car manufacturers with models that are able to charge bidirectionally (includes vehicle-to-load vehicles and vehicles used for testing purposes). For details of specific models, see [35]. Data were gathered in April 2023.

| Manufacturer | Model |
|---|---|
| Abarth | 500e (Cabrio) |
| BYD | Atto 3, Tang, Han |
| Elaris | Beo |
| Fiat | 500e (3 + 1) |
| Fisker | Ocean (Sport, Ultra, Extreme) |
| Ford | F-150 Lighting |
| Genesis | GV60 Sport (Plus), Electrified GV70 Sport, Electrified G80 |
| GMC | Sierra EV Denali Edition 1 |
| Honda | e Advance |
| Hyundai | Ioniq 5, Ioniq 6 |

**Table 3.** *Cont.*

| Manufacturer | Model |
| --- | --- |
| Kia | Niro, EV6 |
| Lexus | RZ 450e AWD |
| Lucid | Air Dream Edition R/P |
| Mercedes | EQE |
| MG | ZS EV, Marvel R |
| Mitsubishi | i-MiEV, Outlander |
| Nio | EL7, ET7 |
| Nissan | e-NV200, Leaf |
| Ora | Funky Cat |
| Polestar | 3 |
| Renault | Megane E-Tech |
| Ssangyon | Koando e-Motion |
| Twike | 5 |
| VW | ID. Buzz Pure/Pro |
| Volvo | EX90 Twin Motor |

**Table 4.** List of charging hardware manufacturers. "X" indicates that the manufacturer offers at least one charging station able to charge through an AC (left) or DC (right) connection to the vehicle.

| Manufacturer | Bidirectional Charging Stations Available | |
| --- | --- | --- |
| | AC | DC |
| ABB | | X |
| Ambibox | | X |
| AME | | X |
| BorgWarner | | X |
| dcbel | X | X |
| Delta | X | X |
| Eaton | | X |
| Enovates | | X |
| Enphase | | X |
| Enteligent | | X |
| Evtex | | X |
| Fermata | | X |
| Ford | | X |
| InCharge | | X |
| Indra | | X |
| Kostal | | X |
| Nichicon | | X |
| Nuvve | | X |
| Silla | | X |
| Wallbox | | X |

Looking at charging stations in Table 4 in contrast, many manufacturers are already able to deliver bidirectional charging stations or have announced such devices. Most suppliers started their portfolio with DC bidirectional charging stations, in line with a general market tendency towards DC for this application. Key reasons for this decision are as follows:

- Lack of EVs with AC bidirectional charging AC bidirectional charging requires the vehicle to perform power delivery in a manner that fulfills grid codes. With limited space in a vehicle, manufacturers initially did not include bidirectional onboard chargers in their vehicles. Without suitable vehicles, there was little motivation for charging station manufacturers to develop AC bidirectional charging stations.
- Easier fulfillment of grid codes The main device that needs to fulfill grid codes is the inverter. If the inverter is stationary, as is the case for DC bidirectional charging, then the inverter can be adapted to the local grid codes. Additionally, this is more similar to traditional grid management strategies where only stationary devices exist that feed power into the grid. In contrast, a vehicle that would potentially move between grid zones is much more challenging to manage if IT systems are set up for stationary devices. For details on this issue, please see Appendix A.
- Limited choice of locations for V2G V2G makes sense if the vehicle can offer its battery energy capacity flexibly [36,37]. This necessarily means that the time that the vehicle is connected to the grid must be longer than what it would take to simply charge a vehicle to the required state of energy. This typically only happens either at home or at work. In consequence, there are typically only one or two sites per vehicle where V2G could occur. This significantly reduces the flexibility that one can derive from having the bidirectional grid-ready inverter inside the vehicle.

### 3.3. Electric Vehicle Sales in Germany

Germany is the lead market by volume in Europe in terms of vehicles sold. For this reason, this paper takes a close look at the market development in the country and implications for V2G applications. In doing so, we do not limit our analysis to vehicles able to charge in the bidirectional mode as this would result in a currently very small number as Table 3 shows. Instead, we aim to understand how other properties of vehicles such as their battery energy capacity and charging power could be applied in a V2G setting.

Figure 1 shows the number of BEVs and PHEVs registered in Germany. It can be seen that from 2020 onwards, the number of cars on the road has approximately doubled each year with the year 2020 showing an increase of 146% in the car stock. Given that the EV car stock in the beginning of 2020 was still comparatively small, the relative growth that year should be seen in this context. In consequence, the potential number of vehicles is steadily increasing.

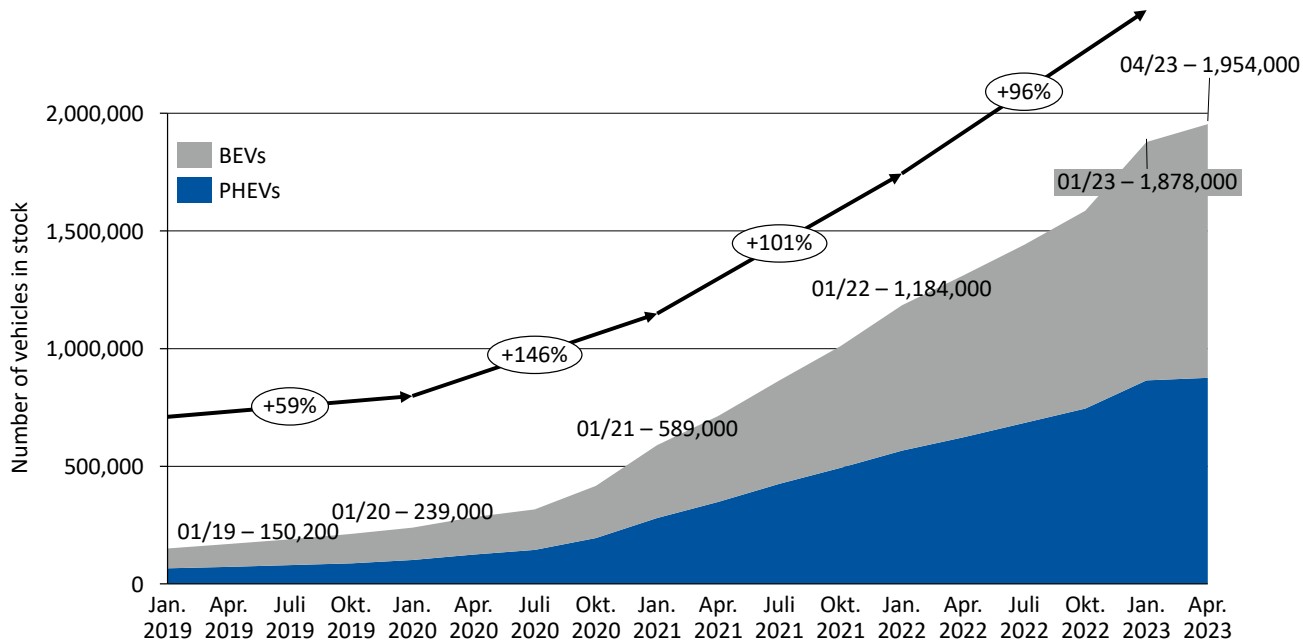

**Figure 1.** Number of BEVs and PHEVs in stock in Germany by quarter [38].

Over the years observed, the share of BEVs in the entire EV fleet as well as the share among new registrations has stayed close to 50%. This has changed in the first months of 2023 where BEVs constituted between 67% (January) and 76% (May) of new registrations. This can be observed in Figure 2. For the same period, however, the overall number of EV registrations dropped significantly. While the beginning of a year has been a period of weak sales over the past years, the scale of the effect was much more pronounced in 2023 as compared to that in previous years. Potential explanations are that subsidies for PHEVs as well as so-called "supercredits" for EVs for EU fleet emission calculations were stopped at the end of 2022 [39]. In terms of V2G applications, a higher overall number of vehicles is of course desirable with the most relevant ones being BEVs as they have a much higher battery energy capacity and much higher charging power per vehicle [23].

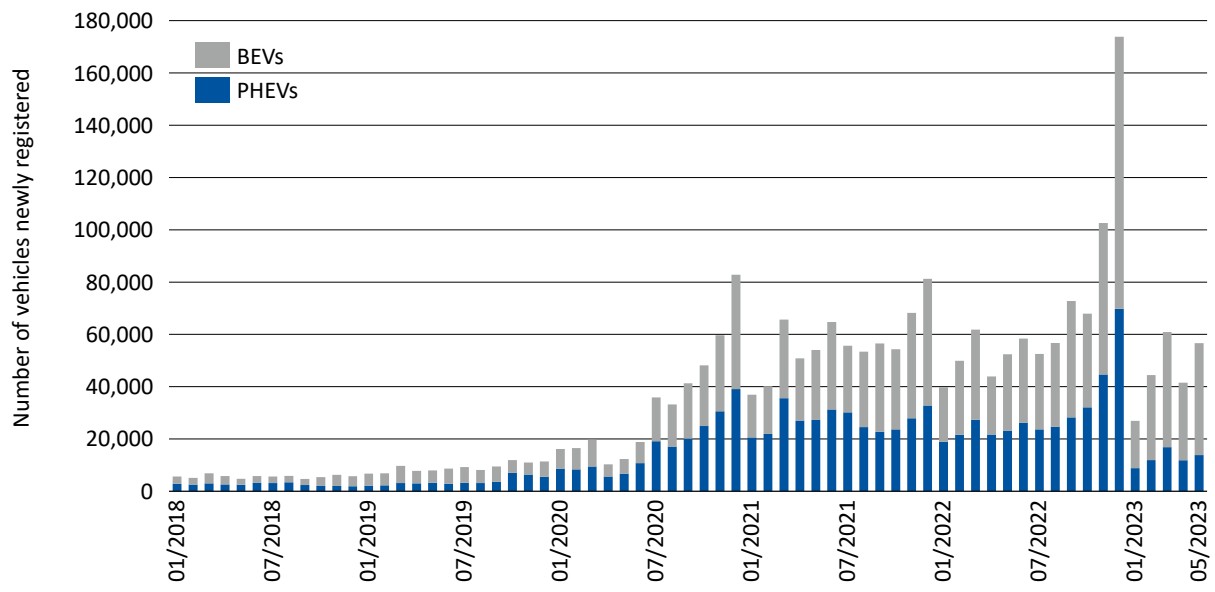

**Figure 2.** Number of newly registered vehicles in Germany per month. Derived from [40] with monthly values based on [41].

### 3.4. Properties of BEVs Sold in Germany

To estimate the market potential for V2G applications, it is necessary to look at the total number of vehicles as well as the properties of those vehicles. In general, vehicles with a larger battery energy capacity are able to offer more flexibility to the grid if we assume that the daily driven distance and consequently the energy consumption are independent of the battery energy capacity. Figure 3 shows the battery energy capacity per vehicle over the time period analyzed in this paper. We can observe a clear increase from around 40 kWh per newly sold vehicle in 2018 to above 60 kWh in 2023. With a typical daily driven distance of 37 km (data from 2017) [42] and an assumed energy consumption of 18 kWh/100 km, approximately 6.7 kWh is needed for the daily commute. With modern vehicles, this leaves over 50 kWh of theoretical flexibility on average—a value that is comparable to that of smaller industrial battery energy storage systems [43]. In practice, flexibility will be reduced since not all vehicles will be connected daily and since users typically desire a minimum battery state of energy so that spontaneous driving is possible.

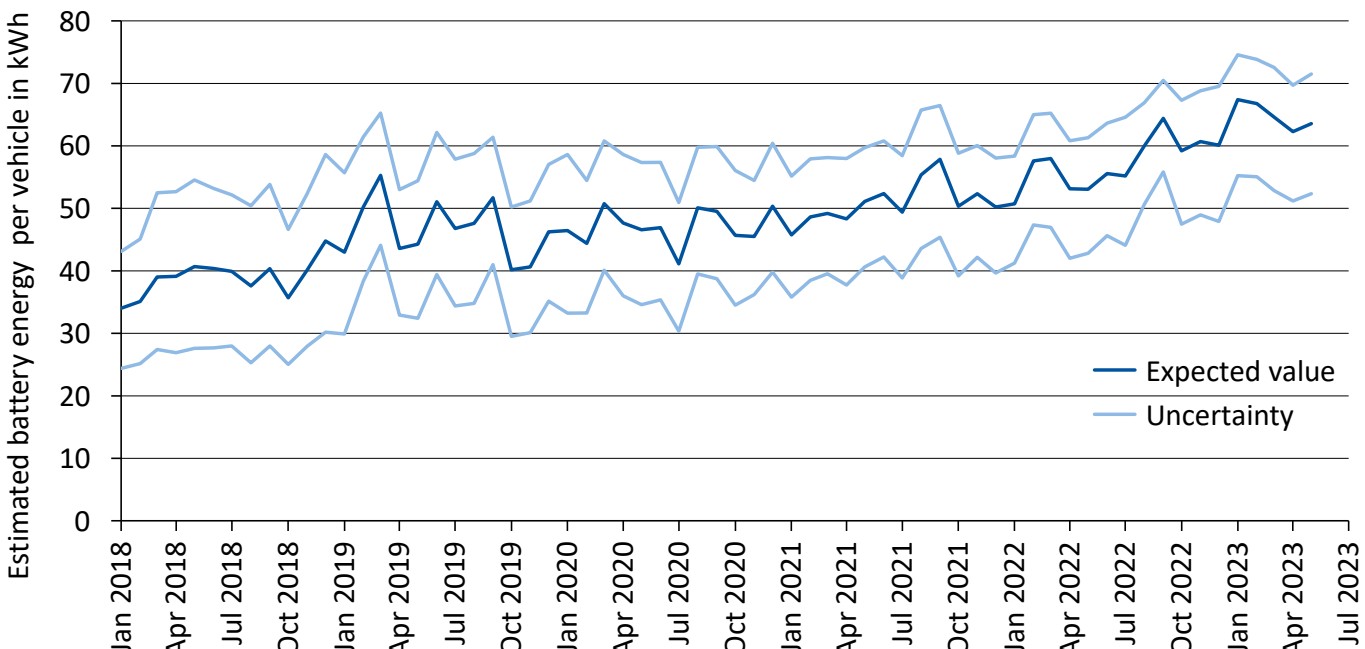

**Figure 3.** Average battery energy per vehicle of BEVs registered per month in Germany. The dark blue line represents the expected value for the battery energy per vehicle while the light blue lines mark the uncertainty window around that value. Derived from [40] with monthly values based on [41] combined with a catalogue of technical vehicle data [27].

In addition to battery energy capacity, charging power and types on connectors are also relevant. Generally, there are two pathways that can be taken here: using an AC connector and having the inverter inside the vehicle or using a DC connector and having the inverter inside a DC charging station. Figure 4 shows the charging power per vehicle for both paths. Although this power is only the charging power, we can assume that discharging powers are likely to be in a similar range. Installed AC capacity has stayed near 8 kW, which is a result of most vehicles being equipped with a three-phase charger able to consume 11 kW and some vehicles having only a single-phase or two-phase port able to consume either 3.7 or 7.4 kW. DC charging powers in turn have grown substantially over time, but with strong differences between the vehicles. One driver for this development is that a DC port was not necessarily installed in a vehicle some years ago while in 2023, over 90% of vehicles are equipped with some DC charging port (see Figure 5). In previous years, these vehicles without a fast-charging port were counted as having a 0 kW connector. In comparison to that of the stationary storage market, the AC charging power per BEV

is larger than that of mean home storage systems and the DC charging powers are in the range of those of larger industrial storage systems [43].

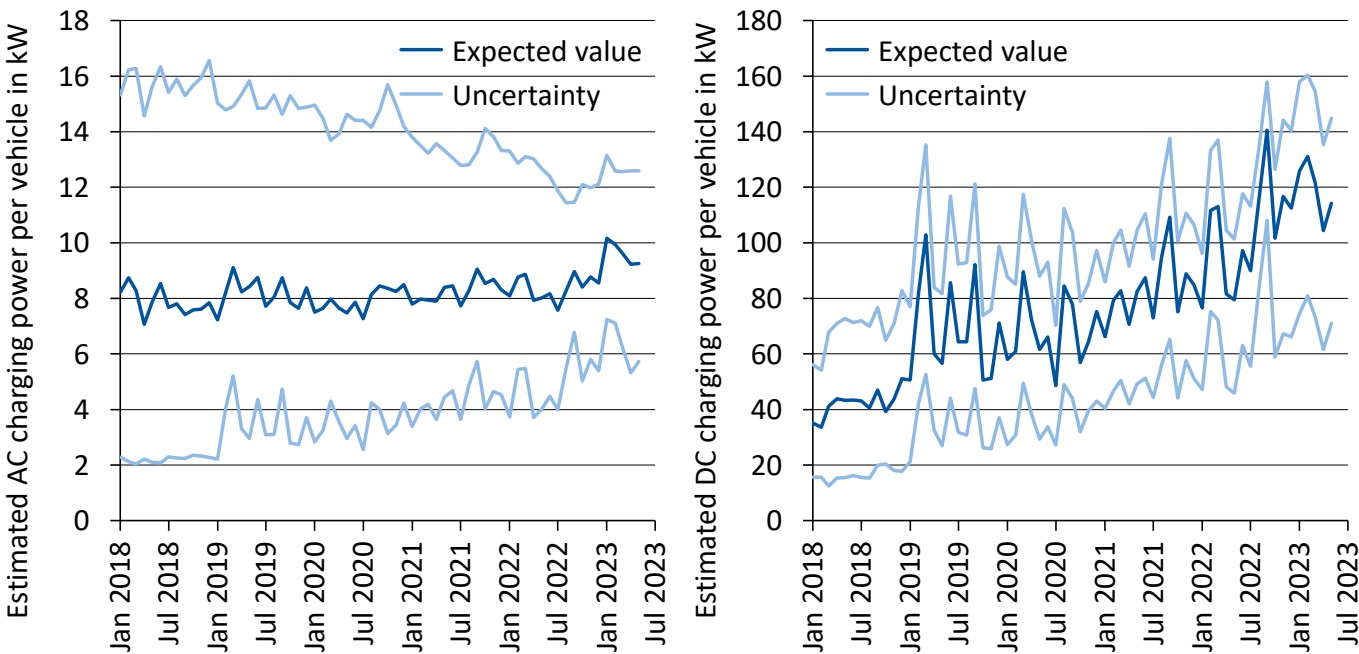

**Figure 4.** Average AC (**left**) and DC (**right**) charging power of newly registered BEVs in Germany. The dark blue line represents the expected value for the charging power per vehicle while the light blue lines mark the uncertainty window around that value. Derived from [40] with monthly values based on [41] combined with a catalogue of technical vehicle data [27]. Note that the light-blue lines show the upper and lower uncertainty boundaries around the estimated value shown in dark blue.

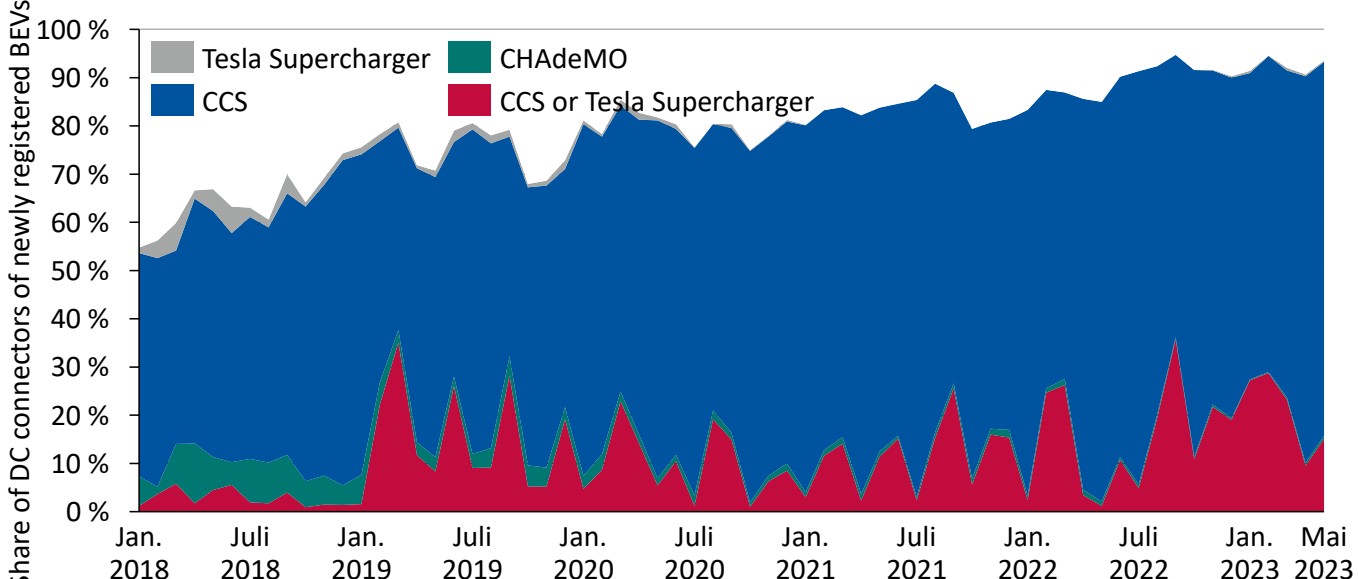

**Figure 5.** Share of DC connectors installed in newly registered vehicles in Germany. Note that the red curve showing CCS or Tesla Supercharger depicts models that were sold with both variants, but that have likely shifted completely to CCS in recent years. The saw tooth pattern is a result of Tesla registering most of its cars in a three-month rhythm. Derived from SOURCE combined with a catalogue of technical vehicle data.

With regard to bidirectional applications, most charging station manufacturers aim for a discharging power of 10 to 20 kW. The flexibility potential in terms of power is consequently comparable for both the AC and the DC path.

The last way of comparing the two paths is to take a look at the connectors used. As Figure 5 shows, over 90% of newly sold vehicles had a CCS connector in 2023. While it has been the dominant standard for some time, CHAdeMO was a used option at the start of the observation period. Particularly, the development of CHAdeMO is relevant for V2G; CHAdeMO was designed to be bidirectional from the beginning and most early field tests with bidirectional vehicles were performed using such vehicles [44]. The ISO 15118-20 norm, in turn, is largely seen as a prerequisite for bidirectional charging for CCS connectors and was only confirmed in April 2022 [30].

## 4. Discussion and Conclusions

This paper discusses some of the key indicators of V2G market readiness in Europe and Germany. The shown statistics provide an overview of which vehicles are ready from a technology point of view, which charging stations are able to operate bidirectionally, and how technical properties of vehicles link to a V2G operation. Given the early stage of the market and the overall rapid development, this paper can only provide a snapshot of the current situation and progress should be expected across all fields quite rapidly. Further, since the market for V2G and bidirectional charging is overall still in its infancy, we decided not to limit the analysis of technical properties of vehicles to models that are bidirectional already. Instead, we analyze features such as battery energy capacity, charging power, and the types of connectors that will become relevant in a V2G system. The graphs shown in this paper should therefore be considered an upper limit and not the real potential.

Even given these limitations, there are several conclusions that we can draw with regard to the V2G market readiness in Germany and Europe:

- The range of vehicles and chargers able to charge in a bidirectional manner is small, but steadily rising. As Tables 1–4 show, the regulatory environment, vehicles, and chargers are beginning to become V2G-enabled, but the options are currently still limited. Additionally, many bidirectional applications are only possible for the connection between specific vehicles and charging stations and in certain countries only. Rapid development can nevertheless be expected.
- The EV market is growing, but 2023 has seen a strong reduction in vehicles sold particularly for PHEVs. While the number of vehicles sold has consistently doubled from the start of 2020 to the end of 2022, this rapid development could not be sustained in the first months of 2023. This effect was particularly strong for PHEVs. For BEVs, the situation is less dramatic, but growth rates have reduced in this field as well.
- BEVs have a large-enough battery energy capacity that V2G is feasible without substantially reducing user comfort. The battery energy capacity of newly registered BEVs has grown to over 60 kWh on average in 2023. This value is much higher than what is required for daily commuting. If a vehicle were able to charge bidirectionally, this excess capacity could be used for bidirectional applications without impacting mobility needs.
- Since virtually all BEVs are equipped with a DC connector, both the AC and DC pathway for bidirectional charging are possible. A key unknown in the move towards bidirectional charging is the role of AC and DC coupling between a vehicle and charging station. Given the technical properties of today's vehicles, no strong indication in either direction is clear. The vast majority of BEVs are equipped with a DC connector meaning that the lack thereof is not a hindrance. Additionally, the discharging power of todays' bidirectional DC charging stations is comparable to the typical AC charging (and in future discharging) power rating.

Future work in this field should separate the market into the relevant segments, vehicle-to-load, vehicle-to-home, and vehicle-to-grid, when discussing the technical properties of vehicles. With the market uptake, such an analysis appears feasible from 2025 onwards.

Additional efforts should be directed towards widening the scope of countries, which necessarily means including diverging data sources from the countries across Europe. Such a study would allow readers to discern trends in the countries of Europe and to allow, especially, countries in early stages of the market to learn from more advanced markets.

**Author Contributions:** Conceptualization, C.H. and J.F.; methodology, C.H.; software, C.H.; validation, C.H. and J.F.; formal analysis, C.H.; investigation, C.H.; resources, C.H., J.F. and D.U.S.; data curation, C.H.; writing—original draft preparation, C.H.; writing—review and editing, J.F.; visualization, C.H.; supervision, J.F. and D.U.S.; project administration, C.H., J.F. and D.U.S.; funding acquisition, C.H., J.F. and D.U.S. All authors have read and agreed to the published version of the manuscript.

**Funding:** This research was funded by the Federal Ministry of Economic Affairs and Climate Actions, grant number 01MV20001A. The project name was "BeNutz LaSA". The literature review of the bidirectional readiness of vehicles, chargers, and countries was performed without external funding.

**Data Availability Statement:** The data used to generate the plots in this paper can be downloaded from https://www.mobility-charts.de/ (accessed on 1 June 2023) free of charge and in the latest available version.

**Acknowledgments:** The literature review for this paper was performed by colleagues at The Mobility House to whom we are grateful.

**Conflicts of Interest:** The authors declare no conflict of interest.

## Appendix A. Grid Code and Communication Protocol Challenges

Grid codes ensure that any device connected to the grid operates in a way that does not deteriorate network quality. Since historically, power plants were predominantly connected to medium- and high-voltage grids, grid codes for power generation plants were focused on these voltage levels. On low-voltage grids, it was typically sufficient to disconnect if the grid reached abnormal states. An example of this behavior is the so-called "49.5 Hz problem" [45]. The problem stems from the fact that until 2015, PV plants, wind turbines, and other renewable generators were required to switch off if the grid frequency dropped below 49.5 Hz. The reasoning behind this logic was that renewable generators had a low generation share at the time and it would be easier to stabilize the system with large and more easily controlled power plants. With the increasing share of renewables, such a behavior could no longer be assumed, since a low frequency already means that there was a generation shortage. Such a sudden disconnect from dozens of gigawatts of power could consequently trigger a blackout. Nowadays, there are requirements for small generators to act in a system-supporting manner by adapting their reactive power flow, provide additional power in the case of frequency dips, etc. [46]. Since EVs in the discharging mode are treated as energy generation plants in many settings, these requirements also apply to them.

In principle, it is possible for EVs to adapt their reactive power consumption or to supply additional power in certain grid situations since power electronic devices can be installed that fulfill these requirements. The challenge is to communicate the desired behavior to vehicles. For a bidirectional DC charging station, this issue is not very critical since the relevant behavior is defined by the charging station. Since the station does not move, the grid codes can be provided to it at the time of installation. For bidirectional AC charging, however, this situation is significantly more challenging; since the vehicle can connect to charging stations in various grid areas, it needs to obtain information on how to behave in unusual grid situations through established communication protocols such as ISO 15118-20, IEC 61851, DIN SPEC 70121, CHAdeMO, GB/T, or NACS. The currently unpublished version of OCPP 2.1 provides data fields to define the necessary behavior [47], but to the best of the authors knowledge, this is not the case for all of the previously stated vehicle-to-station communication protocols. In practice, this means that a vehicle can only provide bidirectional charging services at one (typically home or depot)

charging station. This severely limits the flexibility of using vehicle-to-grid services. To address this, some attempts have already been made such as ISO 15118-20, containing configuration parameter fields for so-called "bidirectional power transfer" including the BPTChannel and GeneratorMode parameter [48]. While these fields already provide some improvements, they do not contain the full information necessary. Similar developments in other standards are likely.

An alternative path to enabling the communication of grid parameters is to unify these across Europe. The EU DSO entity [49] would be a suitable body to enable such a process, but given the major technical differences between the various countries, it appears unlikely that such a path will be followed.

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
