# Peer review of "Vehicle-to-Grid Market Readiness in Europe with a Special Focus on Germany"

_vehicles, doi:10.3390/vehicles5040079_

Round 1

Reviewer 1 Report

The topic is an exciting idea and worthy of a paper in its own right.

- A techno-economic assessment can be performed to show the technological performance and economic feasibility of the V2G technology.

- Existing regulations and grid codes should be investigated in detail to show a road map for this technology. Also, recommendations, key challenges, and future trends related to the improvement of technical requirements for the integration of EVs to the grid can be performed.

- Moreover, grid codes should be investigated in terms of marketing plans and procurement plans. Maybe a pricing roadmap should be recommended in the Electricity Market Law for the V2G technology.

- Auxiliary services that can be provided by the V2G technology can be briefly investigated to show their importance to the readers.

Author Response

The topic is an exciting idea and worthy of a paper in its own right.

- A techno-economic assessment can be performed to show the technological performance and economic feasibility of the V2G technology. -> This topic is one worth of investigation, but other researchers have performed great work in this field as well and we do not see how the data and methodology that we chose for this paper would improve the state of science beyond what has already been done. Instead, we included references to such works in the literature review.

- Existing regulations and grid codes should be investigated in detail to show a road map for this technology. Also, recommendations, key challenges, and future trends related to the improvement of technical requirements for the integration of EVs to the grid can be performed. -> We added a much more extensive discussion on what role grid codes currently play and how that can be improved. Since this is a bit of a niche topic, we decided to move this discussion into the appendix A and kept the very brief discussion in the main body of the text that already existed in the submitted version.

- Moreover, grid codes should be investigated in terms of marketing plans and procurement plans. Maybe a pricing roadmap should be recommended in the Electricity Market Law for the V2G technology. -> We are not quite sure which law you refer to when saying Electricity Market Law. Independent of that, we believe that a proper marketing strategy is a valid goal of a research paper, but outside the scope of our work. We included references in the literature review instead by inserting a section that links to the key publications in describing user behaviour from our point of view. Marketing strategies should be built upon such understandings.

- Auxiliary services that can be provided by the V2G technology can be briefly investigated to show their importance to the readers. -> We included a discussion on which regulatory barrier is most problematic for the two key markets frequency containment reserve and wholesale electricity markets. On the hardware side, including such a discussion would render the presented data unwieldy and we prefer not to include this information for each hardware component.

Reviewer 2 Report

Dear authors,

your paper is very important regarding the upcoming challenges inside the energy and mobility sector.

In the following some information to improve your paper:

- Please check the abbreviation of million in the abstract.

- Keywords: May be you can add "bidirectional charging"

- Line 67/68: You introduce e.g. KBA. Why do you not introduce the abbreviation ADAC?

- Line 125: I think it is better if you give the absolute number of smart meters in Germany and not percentages. 0% looks like there a 0 smart meters in Germany at the end of 2021. According to page 334 and 335 of the report "Monitoringbericht 2022" of the "Bundesnetzagentur" some smart meters were already installed at the end of 2021.

- Line 183: I recommend to start with the subchapter "Electric vehicle Sales in Germany" on p. 6.

- Fig. 1.: I think a bar chart is more pleasant for the readers.

- Please use in all figures the same text size: e.g. ccompare fig. 1. and fig. 3

- Line 220: Which "MiD"-report you use for your analysis? Please add the year in the text or in the references.

Author Response

Dear authors,

your paper is very important regarding the upcoming challenges inside the energy and mobility sector.

In the following some information to improve your paper:

- Please check the abbreviation of million in the abstract. -> Changed

- Keywords: May be you can add "bidirectional charging" -> We thought that this would be covered by vehicle-to-grid already, but added it nevertheless

- Line 67/68: You introduce e.g. KBA. Why do you not introduce the abbreviation ADAC? -> ADAC frequently just uses the abbreviation (similar to companies such as KLM or HP), but we added the full name as well as the translation from which the abbreviation is derived.

- Line 125: I think it is better if you give the absolute number of smart meters in Germany and not percentages. 0% looks like there a 0 smart meters in Germany at the end of 2021. According to page 334 and 335 of the report "Monitoringbericht 2022" of the "Bundesnetzagentur" some smart meters were already installed at the end of 2021. -> We understand that a few smart meters have been installed already by the end of 2021, but this number is generally quite limited and no basis for a scalable business model. We would like to keep the data source for all countries identical and therefore stick to the percentage value given. To highlight more recent changes, we included a reference to a new law allowing private customers to have a smart meter installed for a moderate annual fee. We believe that this information is more relevant when the paper is published than the exact number at the end of 2021.

- Line 183: I recommend to start with the subchapter "Electric vehicle Sales in Germany" on p. 6. -> The first two sections are necessary to understand why we did not only include vehicle-to-grid-ready vehicles in the market analysis. We therefore would prefer to keep the current order.

- Fig. 1.: I think a bar chart is more pleasant for the readers. -> We typically used bar charts for quantities where the current value does not directly depend on the previous value (e.g. for number of new registrations) and area plots or line graphs for values where they do depend on each other (such as the number of vehicles in stock). If you insist, I am happy to change this, but if it is acceptable for to keep it this way given the line of argument just presented, I would prefer to keep our internal style consistent across our various publications.

- Please use in all figures the same text size: e.g. ccompare fig. 1. and fig. 3 -> All figures now have the same font size across all text fields in them. Due to the slightly different dimensions and consequently scaling in the Word / PDF version, they may appear to differ slightly in font size. This will not occur in the web-version of the paper.

- Line 220: Which "MiD"-report you use for your analysis? Please add the year in the text or in the references. -> Unfortunately, the citation format doesn’t allow us to add the year as it is not part of the report title. Instead, we added a small note in the text that indicates 2017 as the year of data.

Reviewer 3 Report

I read with interest your research paper. The topic is interesting and relevant to the ongoing discussion. However, the authors are requested to address the following comments.

1.     TITLE, ABSTRACT AND KEYWORDS: Complete and interesting

2.     INTRODUCTION: Avoid references together. e.g.” have been largely limited to trial projects and small demonstrators [1–4]”.

3.     INTRODUCTION: “the car registrations were retrieved from the German Federal Office for Motor Traffic (KBA) [11] and merged with a catalogue containing technical car data by ADAC [12]” is confusing, e.g. (KBA) ADAC.

4.     INTRODUCTION: Literature review on V2G should be surveyed by involving more recently reported works. Please consider:

https://doi.org/10.1016/j.energy.2023.128354

https://doi.org/10.1016/j.scs.2023.104557

Other relevant work can be included as well and my suggestions are only an example.

Then, I would encourage you to stress brief theoretical and practical implications before the map of the paper. Paragraph 2 deserves more references.

5.     Figure 4 is confusing. Only two of the three curves are explained. Please explain.

6.     Please talk about the future work briefly in the conclusion section

Author Response

I read with interest your research paper. The topic is interesting and relevant to the ongoing discussion. However, the authors are requested to address the following comments.

  1. TITLE, ABSTRACT AND KEYWORDS: Complete and interesting -> Thank you
  2. INTRODUCTION: Avoid references together. e.g.” have been largely limited to trial projects and small demonstrators [1–4]”. -> MDPI states in their Layout file that “The current state of the research field should be carefully reviewed and key publications cited”. References are consequently mandatory.
  3. INTRODUCTION: “the car registrations were retrieved from the German Federal Office for Motor Traffic (KBA) [11] and merged with a catalogue containing technical car data by ADAC [12]” is confusing, e.g. (KBA) ADAC. -> ADAC mostly uses its abbreviation similar to companies like KLM which is why we only used the abbreviated version. We added the full name as per your and another reviewers request.
  4. INTRODUCTION: Literature review on V2G should be surveyed by involving more recently reported works. Please consider:

https://doi.org/10.1016/j.energy.2023.128354

https://doi.org/10.1016/j.scs.2023.104557

Other relevant work can be included as well and my suggestions are only an example.

Then, I would encourage you to stress brief theoretical and practical implications before the map of the paper. Paragraph 2 deserves more references.

We substantially extended the literature review as per your suggestion. This also extends the paragraph 2 that you mentioned that is now embedded in a larger body of literature. We further added a short segment at the end of the first chapter to show what the implications of the paper are.

  1. Figure 4 is confusing. Only two of the three curves are explained. Please explain. -> The lighter shade of blue lines are the uncertainty boundaries (both upper and lower boundary). We added the following sentence to the caption to the caption to hopefully make this more clear: “Note that the light-blue lines show the upper and lower uncertainty boundaries around the esti-mated value shown in dark blue.” If anything else is unclear, please let me know!
  2. Please talk about the future work briefly in the conclusion section -> A paragraph on future work was added.

Round 2

Reviewer 1 Report

The authors addressed all the comments raised by the reviewer. There are no more comments. 

Reviewer 3 Report

All my concerns have been well addressed. I recommend to publish as it is.